# NEUROFABRIC: IDENTIFYING IDEAL TOPOLOGIES FOR TRAINING A PRIORI SPARSE NETWORKS

## ABSTRACT

Long training times of deep neural networks are a bottleneck in machine learning research. The major impediment to fast training is the quadratic growth of both memory and compute requirements of dense and convolutional layers with respect to their information bandwidth. Recently, training 'a priori' sparse networks has been proposed as a method for allowing layers to retain high information bandwidth, while keeping memory and compute low. However, the choice of which sparse topology should be used in these networks is unclear. In this work, we provide a theoretical foundation for the choice of intra-layer topology. First, we derive a new sparse neural network initialization scheme that allows us to explore the space of very deep sparse networks. Next, we evaluate several topologies and show that seemingly similar topologies can often have a large difference in attainable accuracy. To explain these differences, we develop a data-free heuristic that can evaluate a topology independently from the dataset the network will be trained on. We then derive a set of requirements that make a good topology, and arrive at a single topology that satisfies all of them.

## 1 INTRODUCTION

Training deep neural networks requires both powerful hardware and a significant amount of time. Long training times are a significant bottleneck to deep learning research, as researchers typically iteratively design and test new architectures for a specific problem. While a lot of research has been dedicated to accelerating inference, we investigate training as (1) accelerating training can speed up research iteration, (2) evolutionary algorithms for DNN architecture exploration are increasingly being used as an alternative to domain expertise (Jaderberg et al., 2017), and network training is moving to edge devices (Pirk et al., 2019). Unfortunately, the memory requirements of dense, convolutional and recurrent layers grow quadratically with layer information bandwidth [1]. In other words, doubling the size of layer inputs and outputs quadruples the size of the layer. This causes majority of the networks to be memory-bound, making DNN training impractical without batching, a method where training is performed on multiple inputs at a time and updates are aggregated per batch. While batching alleviates the pressure on DRAM bandwidth, it can decrease model accuracy (Masters & Luschi, 2018) especially when scaling training on large clusters (Akiba et al., 2017). Furthermore, larger models in off-chip memory become dominant energy cost (Han et al., 2015a), complicating on-line training on battery-power devices.

Conventional dense and convolutional layers do not offer the user to individually tune layer size and the number of layer inputs and outputs. In this work, we seek a method to decouple the information bandwidth from layer expressivity. Such a method would allow us to (1) speed up training networks by storing them in on-chip memory, (2) remove the memory bottleneck and the need for batching, (3) allow more efficient training on distributed systems, and (4) reduce the energy consumption due to the excessive compute and storage requirements of modern DNNs, potentially allowing us to move training to edge devices. Several works have proposed a priori structured sparsity (Prabhu et al., 2017; Isakov et al., 2018) or weight sharing (Ding et al., 2017) to allow training simpler but 'wider' models. A priori sparsity, where the sparse network topology is selected before training has started, is a promising approach that allows the user to finely and transparently tune the ratio of information bandwidth to memory requirements. If the topology is structured, efficient software

---

[1]We define a layer's information bandwidth as the number of independent signals passing through that layer.

or hardware implementations can be built to accelerate processing with dense network performance . However, before custom architectures or low-level kernels can be built, a general theory of why certain topologies perform – or underperform – is needed. To the best of our knowledge, no work yet tackles the question of the existence of a 'best' topology for sparse neural network training. This paper provides an answer on how a topology should be selected.

Our contributions are as following:

- We propose a sparse cascade architecture that can replace dense or convolutional layers without affecting the rest of the network architecture.
- We develop a sparse neural network initialization scheme that allows us to train very deep sparse networks without suffering from the vanishing gradient effect.
- We evaluate sevaral topologies on a matrix reconstruction task and show that the choice of topology has a strong effect on attainable network accuracy.
- In order to evaluate topologies independently of a dataset, we develop a data-free heuristic for predicting the expressiveness of a given sparse network.
- From the heuristic, we derive requirements that make a good topology, and settle on a single family of sparse networks.

## 2 RELATED WORK

We classify methods that arrive at a sparse network into those that enforce sparsity before, during, or after training. The following is a brief description of each class.

**Enforcing sparsity after training:** In this class of methods, certain weights are zero-ed out after training has finished. This approach has the benefit of first discovering the baseline model accuracy, allowing the training mechanism to evaluate the accuracy to size trade-off. Since training is performed using the dense model, only inference can benefit from these post-training pruning methods. One of the early pruning methods are Optimal Brain Damage (LeCun et al., 1990) and Optimal Brain Surgeon (Hassibi & Stork, 1993), where authors remove weights based on the second derivative of the loss w.r.t. to each weight. The insight here is that removing a weight causes some perturbation in the network, and by picking weights with the smallest second derivative of the loss, the effect of the perturbation on the network functionality is be minimized. DeepCompression (Han et al., 2015a;b) uses a similar approach, but replaces the Hessian-based metric with weight magnitudes. Authors show that high (>95%) sparsity can be achieved as long as networks are finetuned after pruning to restore performance. Alternatively, in (Liu et al., 2015) authors decompose convolutional layers into a set of per-channel basis kernels, which are applied to input feature maps, and a sparse kernel matrix that mixes outputs of the basis kernels into the output feature maps. However, all of these methods lead to unstructured sparsity that is difficult to take advantage of. Structured sparsity, where some assumptions can be made on the structure of sparse kernels, has been explored as a way to improve the execution efficiency of sparse structures on GPUs and CPUs. In (Anwar et al., 2015), authors use particle filters to prune whole channels or kernels. Similarly, Kadetotad et al. explore structured intra-layer sparsity, where instead of individual weights, small blocks of weights are pruned.

**Enforcing sparsity during training:** Instead of pruning after training, pruning can also be applied during training. This has the benefit of potentially reducing the computational load of the training phase, however, the device performing training must still be able to store the whole dense model at the beginning of training. L1 regularization or L1 weight decay is known to cause sparsity during training, as unlike in the case of L2 regularization, all weights will equally be incentivized to approach zero. However, L1 weight decay often causes a decrease in accuracy, and the sparsity is unstructured. In (Wen et al., 2016), authors use Group Lasso (Yuan & Lin, 2006) regularization to enforce the sparsity of more coarse-grained structures instead of individual weights.

**Enforcing sparsity before training:** Model size can also be reduced before training has started. We focus on layer-level methods and not architecture level approaches, as they are orthogonal. Majority of works reducing the size of layers before training have focused either on a priori sparsity or weight reuse. On the weight reuse side, HashedNets (Chen et al., 2015) use a hash function to group multiple weights and have them share and train a single value. CirCNN (Ding et al., 2017) uses block-circulant matrices for storing weights, where elements are shared in a predictable manner and

Fourier transforms are used for inference, reducing the computational complexity of both inference and training.

On the a priori sparsity side, several topologies have been proposed in literature. Deep Expander Networks (X-Nets) (Prabhu et al., 2017) replace dense layers with sparse layers with the expander graph topology. Authors give guarantees of each input neuron being connected to each output neuron within a logarithmic number of layers. Similarly, RadiX-Nets (Robinett & Kepner, 2019) build on X-nets but use the radix topology instead of graph expanders. Alternatively, ClosNets (Isakov et al., 2018) replace a single dense layer with a cascade of three sparse layers with the Clos topology. Clos topology guarantees full connectivity, and has a tunable parameter for the path diversity between all inputs and outputs. While deep expander networks grow in depth with the number of neurons per layer, ClosNets grow in width.

None of the above a priori sparse network works give a definitive answer to which topology maximizes performance per weight. In this work we aim to answer that question.

## 3 APPROACH

The number of parameters in a neural network layer is decided by the number of input and output neurons (in case of fully-connected networks), or the number of input and output channels (in the case of convolutional networks). This prevents decoupling the network bandwidth (i.e. number of inputs or outputs of a certain layer) and the parameter count. We propose that sparsifying layers can allow users to train wider networks without the quadratic growth in network size. Our approach is to replace each fully-connected layer with a cascade of sparsely-connected layers, as shown in Figure 1. The cascade topology and depth are selected at design time so that the number of parameters in the cascade is lower than in the original dense layer. Hidden layer neurons in the cascade have a linear activation function, while the output neurons use the activation of the original network. The cascade needs to have certain properties such as connectivity between all cascade input-output pairs, a small parameter count, and hardware efficiency. In Section 7 we explore the topology requirements further.

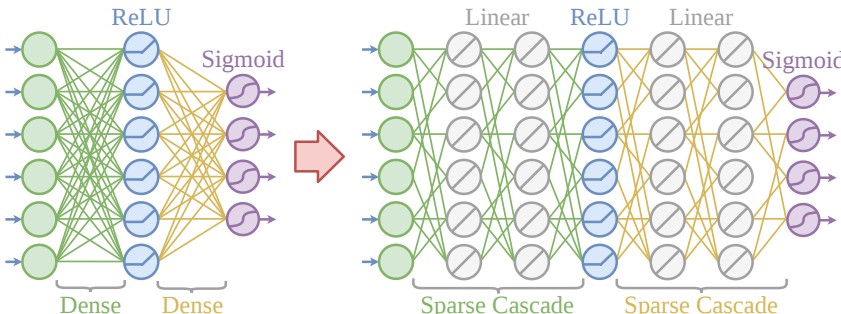

Figure 1: A 2-layer dense network replaced with two 3-layer sparse cascades. Cascades use linear activation functions in their hidden layers, and original activations at their outputs.

Similarly, a priori pruning can be applied to convolutional networks. In conventional CNN layers, each input channel is connected to each output channel by a convolution. For a filter of size $f \times f$, $c$ input and $k$ output channels, the convolutional layer has $f^2ck$ parameters. Since the number of input and output channels $c$ and $k$ directly control both the information bandwidth and the size of the network, we propose to disentangle the number of input/output features and the number of convolutional filters. We adopt the architecture of MobileNets (Howard et al., 2017) and break up convolutional layers into depthwise and pointwise convolutions, as seen in Figure 2a. In the original MobileNets, the majority of parameters belong to the pointwise convolutions, which are simply dense neural networks applied to each 'pixel' individually. We propose to prune only the pointwise convolutions in the same manner we prune dense layers.

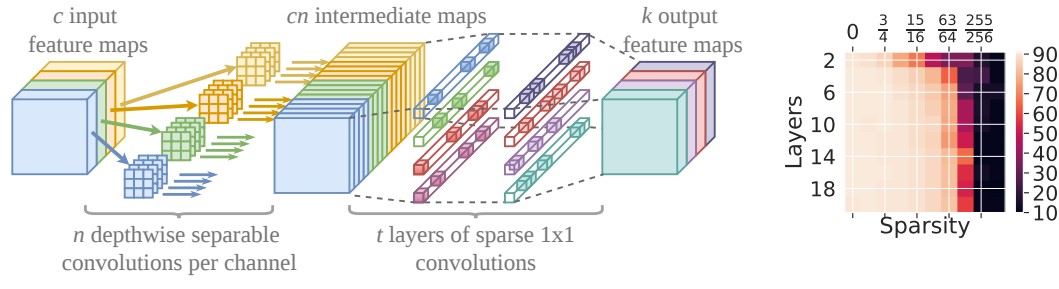

(a) A priori sparse convolutional layer.      (b) Accuracy on MNIST.

Figure 2: (Left) Decomposition of a convolutional layer into a single depthwise and a cascade of sparse pointwise convolutions. (Right) Accuracy of sparse linear networks with varying depth and sparsity on the MNIST dataset.

## 4   INITIALIZING A PRIORI SPARSE NEURAL NETWORKS

Initializing deep neural networks highly affects the training accuracy of the models, but is often not given as much attention as network architecture. If a network is suboptimally initialized, it may train slower than a correctly initialized one or may even not train at all. This problem has been extensively researched in the context of deep neural networks (Glorot & Bengio, 2010; He et al., 2015a) and recurrent neural networks (Henaff et al., 2016). Our proposed approach replaces fully-connected layers with cascades of sparse layers. As such, the new network may be several times deeper than the original one, and may not train as efficiently. This problem is further compounded by the fact that our networks are a priori sparse. Our tests show that deep sparse networks initialized with common initialization schemes like Xavier initalization (Glorot & Bengio, 2010) completely fail to learn. By observing the activation and error values of deep sparse networks, we deduce that these networks suffer from the vanishing gradient problem, i.e., with each successive layer, the variance of both the activations and the errors drops exponentially. In this section, we develop a new initialization scheme for a priori sparse networks that alleviates the vanishing gradient problem by taking layer sparsity into account.

### 4.1   SPARSE XAVIER INITIALIZATION

In Appendix A, we briefly cover the original derivation of the Xavier initialization. Here we generalize it to apply to sparse networks as well. We construct a sparse layer from a matrix $W \in \mathbb{R}^{m \times n}$ by multiplying it element-wise with the mask $M \in \{0, 1\}^{m \times n}$ with sparsity $s \in [0, 1]$. For a random topology, each element $M_{ij}$ of the mask is set as $M_{ij} = Ber(s)$, where $Ber$ is the Bernoulli distribution. For a layer $W_i$ with $n_{in}$ input and $n_{out}$ output neurons, an output neuron's activation variance depends on the variance of each input neuron, each weight connected to it, and the number of input neurons (Appendix Equation 21). For sparse networks, each output neuron is on average only connected to $n_{in}(1-s)$ neurons, hence we update Equation 21 as:

$$\sigma^2(a_{n+1}) = n_{in}(1-s)\sigma^2(a_n)\sigma^2(W_{n+1})$$
$$\sigma^2(\delta^n) = n_{out}(1-s)\sigma^2(\delta_{n+1})\sigma^2(W_{n+1})$$
(1)

Updating the Xavier initialization to take sparsity into account, we write our sparse initialization as:

$$W \sim U\left[ -\frac{\sqrt{6}}{\sqrt{(n_j + n_{j+1})(1-s)}}, \frac{\sqrt{6}}{\sqrt{(n_j + n_{j+1})(1-s)}} \right]$$
(2)

We test the new initialization on the MNIST dataset with networks of different sparsities and depths (Figure 2b). We train randomly-connected networks with 256 neurons in the hidden layers, 1 to 20 hidden layers, and with sparsities between 0 and 255/256. Using the sparse Xavier initialization, we are able to train deep sparse networks. For very sparse networks (sparsity of 63/64 and higher), often there exists no path between certain inputs and outputs, limiting trainability. A better, non-random topology with the same amount of parameters may however be able train.

## 5 TOPOLOGY EXPLORATION

The choice of topology has a strong impact on both the accuracy and the parallelizability of a sparse network. In this section we aim to (1) answer how two topologies can be compared, and (2) create a metric for evaluating a topology independently of a task, given that we can assume nothing about training data beforehand. We first devise a task that allows us to experimentally evaluate a topology. We choose a matrix reconstruction problem where an original matrix $W_O \in \mathbb{R}^{n \times n}$ is reconstructed as a product of $l$ sparse matrices $S_i \in \mathbb{R}^{n \times n}$ with adjacency matrices $M_i \in [0, 1]^{n \times n}$ as:

$$\mathcal{L}(W_O, M_1, ..., M_l) = min \left\| W_O - \prod_{i=1}^{l} (S_i \odot M_i) \right\| \tag{3}$$

The matrix $W_O$ must be random, so that the network cannot abuse any regularities within it. The topology we seek should perform well independently of the matrix structure. Though topologies derived for a specific task may perform better, on average across all tasks, the general topology should achieve the best results. A number of loss functions can be used for this evaluation but we restrict ourselves to L2 loss for now. To gain intuition into the problem, we use the reconstruction loss in Equation 3 on a number of common topologies. Figure 3 illustrates the impact of topology

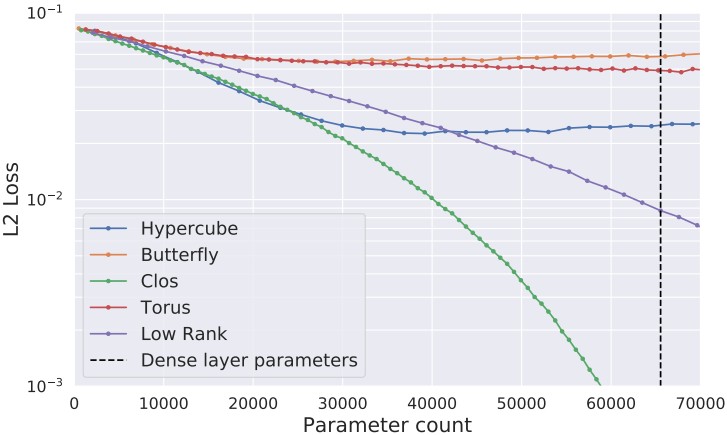

Figure 3: L2 loss of networks with different topologies and varying depths.

choice on overall network accuracy. Our reasoning is that as topologies can underperform on an unseen task, there exists a 'best' topology, one that on average achieves optimal performance. In this and the following section we aim find that topology.

Though we may arrive at an optimal topology purely through evolution, this approach has several issues: (1) evaluating a topology on random matrices is inherently noisy, and we would have to reconstruct many random matrices in order to compare the fitness of two different topologies, (2) the evolved topology is only a point solution, and we would have to rerun the process for a topology with a different number of inputs, outputs, parameters, or layers, and (3) the evolved topologies tell us nothing of the underlying reasons for why a certain topology is underperforming. Therefore, we aim to develop a heuristic that can accurately predict the quality of a topology, so that by analyzing the heuristic we can arrive at the root cause of why certain topologies are underperforming, and produce conclusions on how to construct better performing ones.

### 5.1 L0 CONSTRAINT SATISFACTION

We revisit Equation 3 and define the sparse decomposition $W_d$ as:

$$W_d = \prod_{i=1}^{l} S_i \odot M_i \tag{4}$$

From here we can write an individual element $W_{d\,i,j}$ of matrix $W_d$ from Equation 4 by looking at paths between input $i$ and output $j$ as:

$$W_{d\,i,j} = \sum_p^{P(i,j)} \prod_{e_{mn} \in p}^{p} w_{mn} \tag{5}$$

where $P(i,j)$ is the set of all paths from input neuron $i$ to output neuron $j$. According to Equation 5, in order to satisfy $W_{O\,i,j} = W_{d\,i,j}$, at least one edge in $P(i,j)$ must be set to a specific value. Given an edge $e_{mn}$ with weight $w_{mn}$ that exists on some path between nodes $i$ and $j$:

$$w_{mn} = \left( W_{O\,i,j} - \sum_{p,\,e_{mn} \notin p}^{P(i,j)} \prod_{e_{xy}}^{p} w_{xy} \right) \Big/ \left( \sum_{p,e_{mn} \in p}^{P(i,j)} \prod_{e_{xy}}^{p \backslash e_{mn}} w_{xy} \right) \tag{6}$$

Due to the difficulty of analyzing the quality of topologies using an L2 loss, one approach is to use L0 loss. Here, we task the matrix decomposition $W_d$ with perfectly reconstructing as many individual elements of $W_O$ as possible. The networks are still trained using SGD with L1 loss. After training converges, we can count the number of elements in $W_O - W_d$ where $|W_O - W_d| < \epsilon$ for some arbitrarily small $\epsilon \in \mathbb{R}$. As $w_{mn}$ cannot take multiple values, it can only satisfy a single constraint. Therefore, if $|e|$ is the number of edges in the network, and $|c_{sat}|$ is the number of satisfiable constraints, we can say that $0 \leq |c_{sat}| \leq |e|$. At best, the topology can solve as many constraints as it has edges. As we will show in Section 7, this is not possible to achieve without several modifications to the sparse network.

Equation 6 allows us to reframe the problem of finding the minimal L0 reconstruction loss as a bipartite maximal matching problem. First, we create a set of input-output constraints $\mathbb{C} = \{(i, o) \mid i \in \mathbb{I}, o \in \mathbb{O}\}$, and a set of weights $\mathbb{W} = \{(w_{i,j}^l) \mid M_{l\,i,j} = 1\}$. An edge $w \in \mathbb{W}$ is connected to all constraints $(i, o) \in \mathbb{C}$ where $w \in P(i, o)$. The problem of finding the largest number of elements in $W_O$ that can be perfectly reconstructed by $W_d$ becomes a bipartite maximal matching problem, and can be solved in linear time. However, we find that the maximal number of constraints matched with edges is not a good heuristic for the performance of a topology. Constraint satisfaction counting heuristics fail to account for topology fairness, and equally rate topologies that balance the satisfaction of all input-output pairs, and topologies that greedily satisfy only a subset of input-output-pairs.

## 6 Graph Controllability

In this section, we present a continious method of evaluating the capability of a topology to reconstruct a certain graph. While averaging reconstruction results of many random matrices may provide an estimate of a topology quality, we develop a data-free approach that (1) evaluates a topology in a single pass, and (2) is not sensitive to the randomness of generated matrices.

### 6.1 Neurons need only learn input ratios:

In this work we focus on networks that use ReLU activations between sparse cascades, and linear activations inside them. Both ReLU and linear functions are homogenous, i.e. $f(cx) = cf(x)$ for constants $c \in \mathbb{R}$. Take a neuron with activation $a$ and $n$ inputs with activations and weights $\mathbf{z}, \mathbf{w} \in \mathbb{R}^n$. Activation $a$ can be written as:

$$a = f(\mathbf{w}\mathbf{z}) \tag{7}$$

where $f$ is a homogenous function. We can extract the magnitude and normalize the vector $\mathbf{w}$:

$$a = mf(\mathbf{v}\mathbf{z}), \quad \mathbf{w} = m\mathbf{v}, \quad ||\mathbf{v}||_2^2 = 1 \tag{8}$$

Since the neurons are using homogeneous activation functions, we can shift the job of learning neuron magnitude to the next layer. Now the neuron is only tasked with learning the input ratios.

In the previous section we have used L0 loss to measure the number of constraints a network can solve. Here we see that for $m \times n$ constraints that exist when reconstructing a matrix $W \in \mathbb{R}^{m \times n}$, $n$ of those constraints are magnitude constraints, and $(m-1)n$ are ratio constraints. In other words, a neuron with $m$ inputs has $m-1$ ratio and 1 magnitude constraint. In Appendix B we give a practical way of eliminating magnitude constraints and only measuring the number of ratio constraints.

## 6.2 Neuron control

We define controllability of a certain neuron $n$ w.r.t. to an inputs $a$ and $b$ as the ability of an optimizer to set the ratio of $a$ to $b$ at $n$. We give an inductive definition on three graph primitives: (1) individual input neurons, (2) a neuron with two inputs and one output, and (3) a neuron with one input and two outputs, and show how controllability propagates through them. We show how any sparse neural network can be decomposed into these primitives and how control can be calculated on neurons with larger numbers of inputs and outputs.

**Definition 6.1.** *For a neuron $n$ connected to a set of inputs $\mathbb{I}$, we define the controllability of input $a \in \mathbb{I}$ relative to $b \in \mathbb{I}$ at $n$ as $C_{a/b}(n)$. If $C_{a/b}(n) = 1$, the optimizer can set the ratio $a/b$ at neuron $n$ to any value, without impacting any other ratios already set in the network.*

**Lemma 6.1.** *For inputs $a, b \in \mathbb{I}$ and any neuron $n$, controllability $C_{a/b}(n)$ is bounded as:*

$$0 \leq C_{a/b}(n) + C_{b/a}(n) \leq 1 \tag{9}$$

This is understandable since the optimizer can only set the ratio of inputs $a$ and $b$ at neuron $n$ to a single value, hence controllability of $a$ to $b$ plus controllability of $b$ to $a$ cannot be greater than 1.

**Lemma 6.2.** *For input neurons $a, b, c \in \mathbb{I}$, controllability of ratio $a/b$ at neuron $c$ is:*

$$C_{a/b}(c) = 0 \tag{10}$$

This is obvious since the optimizer has no control over network inputs, and we will use this lemma as the base case of our induction.

**Lemma 6.3.** *For an neuron $n$ connected to a set of inputs $\mathbb{I}$, total controllability of $n$ is limited as:*

$$\sum_a^{\mathbb{I}} \sum_b^{\mathbb{I}} C_{a/b}(n) \leq |\mathbb{I}| - 1 \tag{11}$$

This lemma is a direct result of Section 6.1 limit on the number of ratios.

We now analyze two graph primitives that show how control propagates through the graph.

**Theorem 6.4.** *Control aggregation: For a neuron $n$ with two input neurons $i$ and $j$ connected with weights $w_{in}$ and $w_{jn}$, where $\mathbb{I}_i$ and $\mathbb{I}_j$ are the sets of inputs directly or indirectly connected to neurons $i$ and $j$, the controllability $C_{a/b}(n)$, $a, b \in \mathbb{I}_i \cup \mathbb{I}_j$ is:*

$$C_{a/b}(n) = min(1,\ max(0,\ (C_{a/b}(i) + C_{a/b}(j) + \Delta C_{a/b}(n)))) \tag{12}$$

*where*

$$\sum_a^I \sum_b^I \Delta C_{a/b}(n) \leq \begin{cases} 0, & \text{neither } w_{in} \text{ or } w_{jn} \text{ are tunable} \\ 1, & \text{at least one of the weights is tunable} \end{cases} \tag{13}$$

Intuitively, neuron $n$ inherits the controllabilities of inputs $i$ and $j$, and if at least one weight is tunable, can additionally control the ratio between the two input neurons. This allows it to set an additional ratio between any of the inputs in $\mathbb{I}_i \cup \mathbb{I}_j$. If the loss function is quadratic, instead of using this additional ratio to solve a single constraint, the optimizer may want to partially satisfy multiple constraints. Hence, we allow added controllability $\Delta C_{a/b}$ to have a value in $[0, 1]$. Notice that $\Delta C_{a/b}(n)$: (1) abides by Lemma 6.1, and (2) the optimizer can tune all $|\mathbb{I}_i \cup \mathbb{I}_j|^2$ individual values in $\Delta C$. In corollary C.0.1 we extend this Lemma for the case where $n$ has multiple inputs.

**Theorem 6.5.** *Control fannout: For a neuron $n$ and two output neurons $x$ and $y$ connected with constant connections such that $n = x = y$, $x$ and $y$ controllabilities $C_{a/b}(x)$ and $C_{a/b}(y)$ abide by:*

$$\forall a, b \in \mathbb{I}, \quad C_{a/b}(x) + C_{a/b}(y) = C_{a/b}(n) \tag{14}$$

In other words, neuron $n$'s control is split across outputs. The optimizer chooses how best to split this control, i.e., it does not have to be fair. In Appendix D, we show how graphs can be decomposed so that we can apply Theorems 6.4 and 6.5.

**Corollary 6.5.1.** *For a neuron $n$ with a set of output neurons $\mathbb{O} = \{x_1, ..., x_k\}$ connected with constant connections, output neuron controllabilities $C_{a/b}(x_j)$ abide by:*

$$\forall a, b \in \mathbb{I}, \quad \sum_{x_j}^{\mathbb{O}} C_{a/b}(x_j) = C_{a/b}(n) \tag{15}$$

We can reframe Equation 15 using trainable ratios:

$$\forall a, b \in \mathbb{I}, \; \forall x_j \in \mathbb{O}, \quad C_{a/b}(x_j) = r_{abj}(n) C_{a/b}(n) \tag{16}$$

$$\forall a, b \in \mathbb{I}, \quad \sum_{j=1}^{|\mathbb{O}|} r_{abj}(n) = 1, \quad 0 \le r_{abj}(n) \le 1 \tag{17}$$

## 6.3 Training controllability

Finally, we decompose the sparse cascade's topology so that we can apply Theorems 6.4 and 6.5, in order to write out the equations for the controllability of each output neuron with respect to each input neuron.

Take a cascade with $L$ layers. Each layer has $|n^l|$ sparsely connected neurons, with $|n^0|$ being the number of cascade inputs, and $|n^L|$ being the number of cascade outputs. We define layer $l$ controllability tensor $\mathbf{C}^l \in [0, 1]^{n_0 \times n_0 \times n_l}$ as a tensor where the element $C_{i,j,k}^l$ represents neuron $n_k^l$'s control over ratio $n_i^0 / n_j^0$. The added controllability tensor $\Delta \mathbf{C}^{l+1} \in [0, 1]^{n^0 \times n^0 \times n^{l+1} \times n^l}$ represents the added controllability added by the tunable connections between layer $l$ and $l + 1$, as per Theorem 6.4. The element $\Delta C_{i,j,k,m}^l$ represents the additional control over $i/j$ provided by the edge between $n_m^l$ and $n_k^{l+1}$. Each tensor $\Delta \mathbf{C}_{i,j,:,m}$ abides by Equation 28. The ratio tensor $\mathbf{R}^l \in [0, 1]^{n_0 \times n_0 \times n_{l+1} \times n_l}$ represents the control split from Equation 16 with $R_{i,j,k,m}$ representing the portion of controllability $C_{i/j}(n_m^l)$ passed on to $C_{i/j}(n_k^{l+1})$.

We can propagate controllability through the network as:

$$\mathbf{C}^{l+1} = \mathbf{C}^l \diamond \mathbf{R}^l + \sum_m \Delta C_{i,j,k,m}^l \tag{18}$$

where the $\diamond$ operation is defined as:

$$\diamond : \mathbb{R}^{q \times q \times r} \times \mathbb{R}^{q \times q \times r \times s} \to \mathbb{R}^{q \times q \times s}, \qquad (\mathbf{C} \diamond \mathbf{R})_{i,j,k} = C_{i,j,:} R_{i,j,k,:} \tag{19}$$

The controllability tensor $\mathbf{C}$, added controllability tensor $\Delta \mathbf{C}$ and ratio tensor $\mathbf{R}$ still have to abide by constraints in Lemmas 6.1, 6.2, and Equations 13, 16.

If $n_{in}$ is the number of cascade inputs, we define the controllability loss as:

$$\mathcal{L}(\mathbf{C}^L) = \sum_k \left( (n_{in} - 1) - \sum_i \sum_j C_{i,j,k}^L \right)^2 \tag{20}$$

i.e., if a certain neuron output $k$ has control over $n_{in} - 1$ ratios, that neuron's loss is 0. We can now minimize this loss to discover how much control each cascade output has over the cascade inputs.

## 7 Deriving Better Topologies

In this section, we analyze the results of the controllability heuristic and answer (1) how topologies can be improved on the matrix reconstruction task, and (2) if there exists a definitive answer to what topology performs the best.

### 7.1 The need for skip connections

Similar to skip connections used in ResNet networks (He et al., 2015b), we propose that skip connections in can significantly improve the performance of sparse cascades, though for a different reason.

As mentioned in Section 6.1, if a network uses homogenous activation functions, each neuron only needs to learn the ratio of inputs at that neuron, as the job of learning the magnitude can be shifted to the layer above. Since the number of learnable ratios at a neuron is one less than the number of inputs of the same neuron, that neuron does not need to have all of it's connections trainable. One of the connections can have a constant value of 1, and the network performance will not be impacted. This effect is particularly noticable in butterfly networks, where each neuron only has 2 inputs and 2 outputs, hence 50% of connections are wasted. We replace one connection per neuron with a skip connection valued at 1, and show their performance in Figure 4. See that skip connections significantly improve the performance of butterfly and hypercube networks.

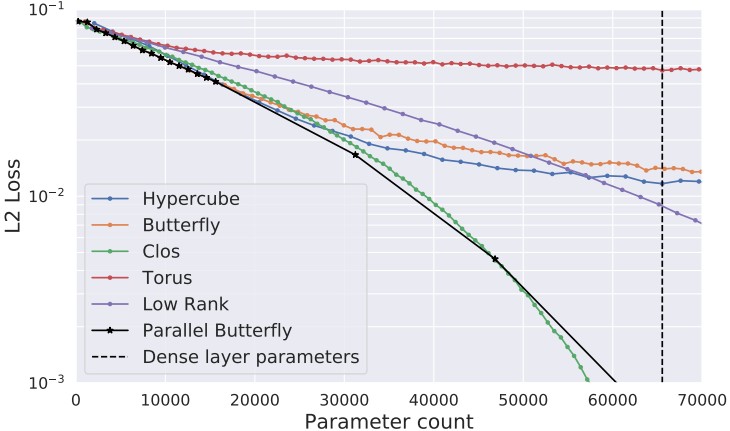

Figure 4: L2 loss of networks using skip connections

## 7.2 THE NEED FOR INPUT-OUTPUT PAIR EQUALITY

While skip connections help topologies achieve similar performance (see hypercube and butterfly in Figure 4), some topologies still outperform / underperform. We turn to our controllability heuristic for an explanation of this behavior. We train our controllability network from Section 6.3 with topologies from Figure 4. The trained network produces $C^L$, the controllability tensor of the last layer of the network. This is a 3D tensor where $C_{i,j,k}^L$ specifies the optimizer's control of input ratio $n_i^0/n_j^0$ at output neuron $n_k^L$. Since we optimize for the number of ratios set, and not the specific configuration, we sum $C^L$ in the second dimension with $K_{i,k}^L = \sum C_{i,:,k}^L$. We plot the resulting $K$ matrices in Figure 5 (a-d).

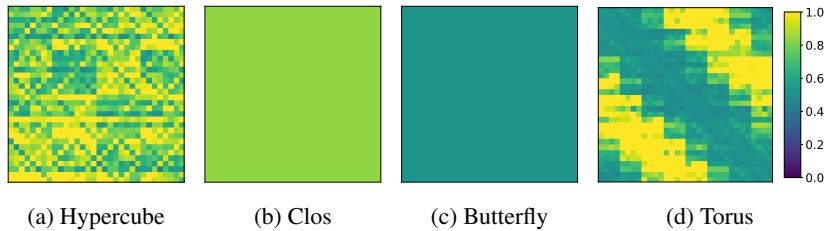

(a) Hypercube      (b) Clos      (c) Butterfly      (d) Torus

Figure 5: Controllability matrix $K$ after 1000 training iterations. All networks have 32 inputs and outputs, and 1152, 1152, 1152, and 1120 edges, respectively. Clos router configuration is (8, 9, 8), hypercube has 6 layers, butterfly has 18 layers, and the torus has 8 rows and 4 columns.

The total controllability a network achieves is equal to the sum of $K$, and can be interpreted as the total 'brightness' of Figures 5 (a-d). With skip connections and a topology that does not oversaturate certain input-output pairs, we can guarantee that the number of controllable ratios is equal to the number of trainable edges in the graph. Notice that the hypercube and torus figures have significant variance, while Clos and butterfly networks are smooth. This means Clos and butterfly networks do not prioritize any specific input-output pair. Since torii and hypercubes are examples of small-world networks (Watts & Strogatz, 1998), these networks find it easier to satisfy closer input-output pairs. Even though these networks are trained with L2 loss, where outlier pairs are incentivized

to approach the mean, torus and hypercube networks show signifcant vraiance. This is important since when given a matrix $W_O$ which is to be reconstructed, permuting $W_O$ columns or rows in a targeted way may improve reconstruction accuracy. Ideally, the position of an input within the topology should not impact the performance of the network. In Figure 5, we give four examples: hypercubes whose controllability matrix has high mean and high variance, Clos with high mean and low variance, butterfly with low mean and low variance, and torus with low mean and high variance.

### 7.3 THE NEED FOR SHALLOWNESS

Saxe et al. (2013) explore the dynamics of training deep linear networks, and show that deep linear networks have at most a constant time slower convergence compared to shallow linear networks. This effect (1) may still be detrimental to training, and (2) to the best of our knowledge, has not been studied for the case of sparse deep linear networks. Hence, when choosing between two otherwise equivalent topologies (i.e., topologies with the same controllability mean and variance), we should choose the shallower one. Furthermore, observing Figure 3, we see that after a certain depth, butterfly, torus, and hypercube networks lose performance with depth, despite gaining parameters. This is likely due to an issue with initializing very sparse deep networks, as sparse random initializations may be more vulnerable to noise compared to dense networks. On the other hand, constant-depth topologies such as Clos (with depth 3) and low rank (with depth 2) eventually outperform all other variable-depth topologies. Similarly, our ideal topology should have constant depth.

### 7.4 THE NEED FOR HIGH INFORMATION BANDWIDTH

In Figure 4 we notice that for small parameter counts, the Clos topology is outperformed by both butterflies and hypercubes. By analyzing the controllability matrices of low-parameter Clos networks, we see that this behavior stems from the limited information bandwidth of Clos networks. In Figure 8a, we show an example of a Clos network that underperforms due to limited bandwidth.

### 7.5 ONE TOPOLOGY TO RULE THEM ALL

We evaluate different topologies with the above criterions, namely: (1) a topology should use skip connections, (2) the controllability matrix of a topology should have no variance, (3) the topology depth should not change with the number of parameters, and (4) the topology should have high information bandwidth, independent of the number of parameters. All of the above topologies can satisfy constraint (1) given skip connections, however, only Clos and butterfly satisfy constraint (2). Since butterfly, hypercube and torus topologies grow in depth as the parameter budget grows, while Clos grows in width, only Clos satisfies constraint (3). However, while butterfly, hypercube, and torus satisfy requirement (4), Clos does not. Hence, we propose a topology we call *parallel butterfly*, which satisfies all of these constraints. Parallel butterfly consists $p$ butterfly networks of maximum depth $d$, where $d$ is a metaparameter selected ahead of time. All $p$ networks are connected to the same inputs, and sum their outputs. With a small number of parameters at it's disposal, the parallel butterfly sets $p = 1$, and grows in depth up to $d$ layers. Afterwards, it grows in width by increasing the parameter $p$. In Figure 4 we show that parallel butterfly outperforms all other topologies. In Appendix Figure 8b, we give an example of a parallel butterfly topology where $p = 2$.

## 8 CONCLUSION

In this work, we have explored accelerating DNN training by pruning networks ahead of time. We proposed replacing dense and convolutional layers using sparse cascades with topologies selected ahead of time. We presented an a priori sparse neural network initialization scheme that allows us to train very deep networks without the vanishing gradient problem. Since networks are pruned before the model has seen any training data, we investigated topologies that maximize accuracy over any domain. We have developed a data-free heuristic that can evaluate the sparse network's control of outputs with respect to inputs, allowing us to assess the expressiveness of a given topology. We have extracted several requirements that make for a good topology, such as the need for skip connections, information bandwidth, shallowness, and input-output pair equality. Finally, we have proposed a topology we call *parallel butterfly* as the ideal topology for training a priori sparse networks, and have experimentally shown that it outperforms other considered topologies.

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

## A  INITIALIZING DEEP LINEAR NEURAL NETWORKS

In (Glorot & Bengio, 2010), authors propose that the difficulty of training deep neural networks lies in their initialization. They observe that for the common weight initialization of $W = U[-\frac{1}{\sqrt{n}}, \frac{1}{\sqrt{n}}]$, the variance of activations decreases as the signal progresses through the layers. Similarly, the variance of the gradients is the highest at the last layer, and decreases as gradients are backpropagated towards the input layer. We briefly cover a derivation of the Xavier initialization here.

Given a layer with $n_{in}$ input and $n_{out}$ output neurons, and a uniform element-wise variance $\sigma^2(a_i)$ of $i$-th layer activations, $\sigma^2(W_i)$ of $i$-th layer weights, and $\sigma^2(\delta_i)$ of $i$-th layer gradients, we can calculate the variance of the next / previous layer's actvations and gradients as:

$$\sigma^2(a_{n+1}) = n_{in}\sigma^2(a_n)\sigma^2(W_{n+1})$$
$$\sigma^2(\delta^n) = n_{out}\sigma^2(\delta_{n+1})\sigma^2(W_{n+1})$$
$$(21)$$

In order to maintain the variance accross layers, layer $i$ and $i + 1$ activation / gradient variances should be equal:

$$\sigma^2(a_n) = \sigma^2(a_{n+1}) \implies \sigma^2(W_{n+1}) = \frac{1}{n_{in}}$$
$$\sigma^2(\delta_n) = \sigma^2(\delta_{n+1}) \implies \sigma^2(W_{n+1}) = \frac{1}{n_{out}}$$
$$(22)$$

For non-square weight matrices, authors compromise and set the weight variance as:

$$\sigma^2(W_{n+1}) = \frac{2}{n_{in} + n_{out}} \tag{23}$$

If the weight matrix is initialized with a uniform distribution $W = U(-r, r)$, the distribution variance can be calculated as:

$$\sigma^2(U(-r, r)) = \frac{r^2}{3} \tag{24}$$

From equations 23 and 24 we have:

$$\frac{2}{n_{in} + n_{out}} = \frac{r2}{3}$$
$$r = \frac{\sqrt{6}}{\sqrt{n_{in} + n_{out}}}$$

(25)

Weights should then be initialized with the following distribution, commonly known as the Xavier initialization:

$$W \sim U\left[-\frac{\sqrt{6}}{\sqrt{n_{in} + n_{out}}}, \frac{\sqrt{6}}{\sqrt{n_{in} + n_{out}}}\right]$$

(26)

## B    MEASURING THE NUMBER OF SOLVABLE RATIO CONSTRAINTS

On a practical note, one way to test how many ratios a network can learn is to append a 'diagonal layer' to the end of the network (i.e., a new layer with a single neuron attached to each output), as seen in Figure 6. The diagonal layer is a diagonal matrix whose only trainable elements are on the main diagonal, and all other values are 0. When training a network, this diagonal layer can only learn magnitudes, and not ratios between signals, because each neuron only has one input and cannot 'mix' any signals. This gives us an easy way of measuring the number of ratios a network can correctly express: we train a network with L1 loss until it converges. We then count the number of constraints $k$ the network has satisfied. These constraints can be ratio constraints or magnitude constraints. If we have $n$ output neurons, we know that the last layer will have satisfied all magnitude constraints. Hence, the number of ratios the network can satisfy is $k - n$. For example, the network in Figure 6 (right, though true for left too) can satisfy three out of the 4 absolute constraints. 2 of those are magnitude constraints, meaning it can only satisfy one ratio constraint. That ratio is calculated at neuron $n$, so either neuron $x$ or $y$ can get a correct ratio of inputs, but not both. Of course, with L2 loss, the network will settle for a solution that doesn't satisfy either, but picks some middle ground.

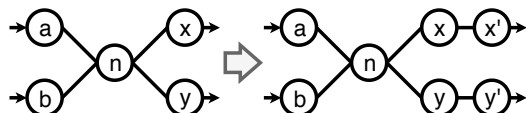

Figure 6: A 'diagonal layer' allows networks to solve magnitude constraints even if the network uses constant (non-tunable) connections in the last layer.

## C    CONTROLLABILITY COROLLARIES

**Corollary C.0.1.** *For a neuron $n$ with a set of input neurons $\{i_1, ..., i_k\}$ connected with $t$ trainable and $k - t$ constant connections, controllability $C_{a/b}(n)$ is:*

$$C_{a/b}(n) = min(1, max(0, \sum_{j}^{k} C_{a/b}(i_j) + \Delta C_{a/b}(n)))$$

(27)

*where*

$$\sum_{a}^{\mathbb{I}} \sum_{b}^{\mathbb{I}} \Delta C_{a/b}(n) \leq min(t, k-1)$$

(28)

*Notice that as at least one connection is constant, the network can make full use of all trainable connections.*

## D    DECOMPOSING GRAPHS

We briefly show how graphs can be decomposed so that we can apply Theorems 6.4 and 6.5. In Figure 7 we see a 3-1-2 graph decomposed into a graph where each neuron has at most 2 inputs

or outputs. We can apply Theorem 6.4 to subgraph $\{a, b, m\}$ and $\{m, c, n\}$, and Theorem 6.5 to subgraph $\{n, x', y'\}$. Since neurons $x$ and $y$ only have one input, their controllability is identical to that of $x'$ and $y'$, respectively.

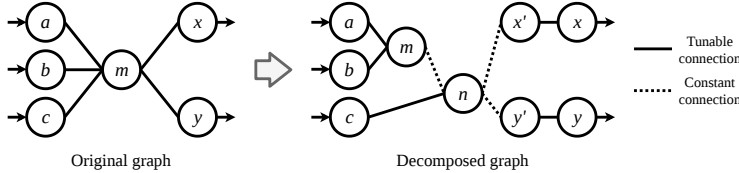

Figure 7: Decomposition of a 3-1-2 graph into a new graph on which we can apply aggregation and fannout theorems.

# E  EXAMPLE TOPOLOGIES

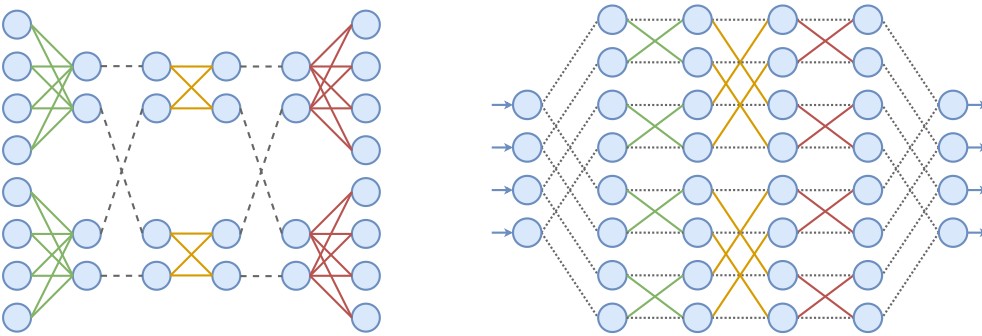

(a) Example of a Clos network that underperforms due to limited bandwidth in the middle layer. Dotted lines represent constant-valued connections.

(b) Example of a parallel butterfly topology with two 4-input, 3-layer butterfly topologies in parallel. Dotted lines represent constant-valued connections.

Figure 8: Example topologies

