# OpenReview forum: "NeuroFabric: Identifying Ideal Topologies for Training A Priori Sparse Networks"
_ICLR.cc/2020/Conference — Reject_

### Official Review · AnonReviewer3 · 2019-10-17
**Official Blind Review #3**

**Rating:** 3

**Review:**

This paper tackles the problem of finding a sparse network architecture before training, so as to facilitate training on resource-constrained platforms. To this end, the authors propose to replace dense layers with series of sparsely-connected linear layers. They then study how to initialize such sparse layers to avoid gradient vanishing. Furthermore, they propose an approach to finding the best topology by measuring how well the sparse layers can approximate random weights of their dense counterparts.

Methodology/clarity:
While I found the beginning of the paper, up to Section 4 (included), easy to follow, I must admit that I have been struggling with the remainder of the paper. In particular:
- In Section 5, the authors propose to use the L2 difference between random weights of a dense layer and the reconstruction of these weights with sparse layers. It seems that the author take this as a proxy to network accuracy (as indicated by a statement just below Fig. 3). It is not clear to me, however, that a small L2 norm will indeed correspond to a similar accuracy in practice, and this is never demonstrated in the paper.
- This L2 loss is then replaced by an L0 one, which again I fail to find any justification for. In fact, I don't see why the authors did not directly use the L0 norm in the first place.
- If I understood correctly, the goal of Sections 5-6 is to find the best topology for the sparse layers. However, it seems the number and width of such layers still need to be pre-defined. Correct?
- Below Eq. 6, the authors mention that the networks are still trained using SGD with L1 loss. I do not understand this statement. On what is the L1 loss computed?

Experiments:
Ultimately, the authors argue that their approach allowed them to find a parallel butterfly architecture that outperforms the other ones. However:
- There is no evidence that the resulting architecture is indeed better than the others when it comes to solving an actual problem (e.g., in terms of classification accuracy);
- There still seem to be a fair bit of manual design in this parallel butterfly architecture, and it is therefore not clear to me that it is truly the best possible one.

Related work:
A number of architecture designs have been proposed to obtain a compact network prior to training. MobileNets, used here as baseline CNN, is one of them, but so are MobileNetv2, ShuffleNet and ShuffleNetv2, SqueezeNet, and ERFNet. I believe that it would be worth discussing these architectures and showing the benefits of the proposed approach over them.

Summary:
I have been struggling to follow the second half of this paper, and I am not convinced by the hypothesis that the L2 (or L0) norm between dense and sparse parameters is a good proxy for network accuracy, which is not demonstrated. I therefore feel that this paper is not ready for publication.


**Experience Assessment:**

I have published one or two papers in this area.

**Review Assessment: Checking Correctness Of Derivations And Theory:**

I assessed the sensibility of the derivations and theory.

**Review Assessment: Checking Correctness Of Experiments:**

I assessed the sensibility of the experiments.

**Review Assessment: Thoroughness In Paper Reading:**

I read the paper thoroughly.

---

> ### Author Response · Authors · 2019-11-15
> **Review #3 response**
>
> We thank the reviewer for the extensive review and very insightful comments. We hope to address all of the
> concerns in this response and the next iteration of the document.
>
> Concerning whether L2 reconstruction loss is a good proxy of the final accuracy, we agree with the reviewer
> that this is an unanswered question. As of now, we only have empirical evidence that decreasing L2
> reconstruction loss leads to better network performance. In the updated version of the paper, we will add
> several benchmarks (CIFAR10, ImageNet, etc.) and compare the L2 reconstruction loss of a given set of networks
> with the accuracy on real problems. In the future we hope to have theoretical evidence supporting this claim.
>
> Concerning L0 norm, we initially used it for measuring the reconstruction quality of networks trained with L1
> loss, where the network will either perfectly reconstruct certain values (loss of 0), or have some error (loss
> of 1, independent of actual error). However, we discovered that L0 measure leads to topologies that will
> reconstruct a number of matrix elements perfectly, yet the topology will underperform on real tasks. This is
> due to the fact that L0 metric does not try to 'balance out' the reconstruction quality over the whole matrix,
> but instead is satisfied with greedily reconstructing some subset of elements. In the revision, we will clear
> up why L0 loss is not a good proxy for real world performance of a sparse topology.
>
> Concerning the question on whether the number and width of layers needs to be predefined: our work mainly aims
> to 'retrofit' existing neural network architectures. As such, the types of layers and layer width are already
> provided by the network designer, and our work only attempts to break down the already specified linear layers.
> The depth of the cascade does however have to be manually selected, but as it depends on the parameter budget
> the user has provided, it can be calculated from the layer width and number of parameters. We now understand
> that we should clear up these points raised by the reviewer.
>
> Concerning training with L1 loss, what we mean is that the networks are trained on the matrix reconstruction
> task using L1 loss, but are finally evaluated using L0 loss. We will clear up this point in the updated version
> of the document.
>
> Finally, we agree with the reviewer on the experimental points, and will update the paper with experimental
> results on CIFAR10, ImageNet, and more. For the second point, we will update the paper with more experimental
> evidence. We plan to add an experiment using neuroevolution, where we will compare evolved topologies to our
> predicted ones, on both matrix reconstruction and real world tasks.
>
> Concerning other compact architectures the reviewer mentioned (MobileNetv2, ShuffleNet and ShuffleNetv2,
> SqueezeNet, and ERFNet), we plan to apply our method to these architectures and evaluate whether further memory reductions can be gained, and at what cost to accuracy.
>
> Again, we thank the reviewer for the time and effort put into helping us better this paper.

---

### Official Review · AnonReviewer2 · 2019-10-25
**Official Blind Review #2**

**Rating:** 3

**Review:**

The paper proposes a sparse cascade architecture that is a multiplication of several sparse matrices. Then the paper provides several considerations about the connectivity scheme. Finally, the paper proposes a specific connectivity pattern that outperforms other ones.

I am not exactly in this area, but the paper is frequently confusing and hard to read. If I understand correctly, Sec 5 proposes a method to evaluate topologies. It seems that this method is strongly tied to the proposed cascade architecture, therefore cannot compare cascade vs non-cascade. The choice of the matrix reconstruction problem looks arbitrary. The paper finishes very abruptly.

Another consideration is that evaluating on random matrices doesn't demonstrate the ability to solve real world problems. There are no experimentation on real benchmarks or comparison to prior work on sparse networks. It should be possible to compare to methods that sparsify networks after training as well as to methods that enforce sparsity during/before training.

Other:
- it is strange to call a method by the first name of the researcher (Xavier)
- there are some grammatical mistakes and problems with articles

EDIT:
After the rebuttal period paper still has weak experimentation. My score stays the same.

**Experience Assessment:**

I do not know much about this area.

**Review Assessment: Checking Correctness Of Derivations And Theory:**

I assessed the sensibility of the derivations and theory.

**Review Assessment: Checking Correctness Of Experiments:**

N/A

**Review Assessment: Thoroughness In Paper Reading:**

I read the paper at least twice and used my best judgement in assessing the paper.

---

> ### Author Response · Authors · 2019-11-15
> **Review #2 response**
>
> We thank the reviewer for their effort and comments.
>
> We understand that the motivation for using matrix reconstruction as a proxy for evaluating network
> performance is somewhat unclear. In future iterations of the paper we will strengthen our argument for
> why using a matrix reconstruction task is an appropriate measure for evaluating sparse topologies.
>
> Also, we plan to add several benchmarks to the appendix, e.g., CIFAR and ImageNet evaluations.
>
> Concerning the Xavier/Glorot initialization, both the author's first and last name have been used in
> literature. We have adopted the more common (at least in our experience) naming.

---

### Official Review · AnonReviewer1 · 2019-10-30
**Official Blind Review #1**

**Rating:** 3

**Review:**

The authors propose a new weight initialization method for sparse neural networks and develop a weight topology that satisfies desirable properties. Their derivation is data-free, and thus the analysis should generalize to arbitrary datasets. They demonstrate that their new topology outperforms existing approaches on a matrix reconstruction task.

Overall I think this work is an interesting direction for designing static sparse neural network weight topologies, but it’s lacking in empirical evidence of their claims and could do better to tie themselves to existing literature in training sparse neural networks.

If the authors could strengthen their results by a) experimenting with their newfound topology and initialization on standard sparsification benchmarks like CIFAR, ImageNet, and WMT EnDe b) comparing their approach to other static-sparse [1, 2, 3] and dynamic-sparse [4, 5] training algorithms this could be a good paper, but without more experimentation it’s unclear what can be taken away from this work. If the authors added results in this direction I would be willing to increase my score.

Comments on Claims of the Paper:

1. “Cascades” are a known trick in both dense & sparse neural networks [2, 6].
2. The authors describe their motivation for developing a new sparse initialization method in the first paragraph of section 4. It would be nice to see some of this experimental data, such that we could understand the magnitude of the vanishing gradient problem in these tests and see that the newly derived initialization alleviates it. The reason for my skepticism is that Liu et al [7] used a similar scheme where they re-scale the standard deviation of the gaussian based on the fraction of nonzero weights, but later found that it made no difference in their results for unstructured pruning (which I learned through discussion with the authors).
4. The data-free derivation approach makes sense and I understand that this makes the approach theoretically applicable to arbitrary datasets, but the authors do not apply it to other datasets to show that it generalizes in practice.
5. The authors show that their topology outperforms on the matrix reconstruction task, but they don’t compare with other sparsification approaches used in deep learning like sparse evolutionary training [4], or SNIP [1] (note that these techniques also maintain a sparse network during training, as opposed to pruning approaches like magnitude pruning [8] that are dense during training but sparse for inference).

Comments on the Results of the Paper:

The authors appear to contextualize their work in deep neural networks, but all of their experimental results are on matrix reconstruction or linear models on MNIST. This is sufficient for analysis and motivation, but taking the developed approaches and applying them to a deep neural network and showing improvements would go a long way towards improving this paper.

Assorted Notes:

In the first paragraph of the introduction:

“In other words, doubling the size of layer inputs and outputs quadruples the size of the layer. This causes majority of the networks to be memory-bound, making DNN training impractical without batching, a method where training is performed on multiple inputs at a time and updates are aggregated per batch.”

It is correct that a matrix-vector product (i.e., neural network training with batch size 1) is typically memory bound on accelerators like GPUs, but it’s not clear why the quadratic growth in computational cost with input/output size has anything to do with this. The cause of memory bound-ness is lack of reuse of operands, which can be alleviated by increasing the batch size s.t. the computation becomes a matrix-matrix product. Batch size 1 training is also not desirable. Recent work has shown that large batch sizes do not degrade model quality with proper hyperparameter tuning [9], and larger batch sizes are desirable from a hardware perspective to achieve higher throughput.

References:
1. https://arxiv.org/abs/1810.02340
2. https://d4mucfpksywv.cloudfront.net/blocksparse/blocksparsepaper.pdf
3. https://arxiv.org/abs/1903.05895
4. https://www.nature.com/articles/s41467-018-04316-3
5. https://openreview.net/forum?id=ryg7vA4tPB
6. https://arxiv.org/abs/1811.10495v3
7. https://arxiv.org/pdf/1810.05270v2.pdf
8. https://arxiv.org/abs/1710.01878
9. https://arxiv.org/abs/1811.03600

**Experience Assessment:**

I have published one or two papers in this area.

**Review Assessment: Checking Correctness Of Derivations And Theory:**

I assessed the sensibility of the derivations and theory.

**Review Assessment: Checking Correctness Of Experiments:**

I carefully checked the experiments.

**Review Assessment: Thoroughness In Paper Reading:**

I read the paper thoroughly.

---

> ### Author Response · Authors · 2019-11-15
> **Review #1 response**
>
> We thank the reviewer for the extensive review and helpful comments. We hope to address all of them in
> this response and the next version of the paper.
>
> Concerning the vanishing gradient problem, in the next iteration of the paper we will update the appendix
> with some empirical results showing the activation magnitude w.r.t. sparsity and depth. Our experiments
> have shown that sparsity-aware initialization is crucial for training deep networks. As mentioned in the paper,
> if given a layer density $d$, by using original Xavier initialization, we expect that each layer's
> activation variance is $d$ times lower than that of the previous layer, causing an exponential drop and
> preventing training with only several sparse layers. In the updated version we will show a comparison of
> layer activation variances with varying depths and sparsities.
>
> Concerning the lack of empirical data on real world datasets, we will update the next version of the paper
> with experiments on real world datasets.
>
> For the last point, we agree with the reviewer, and will compare our approach to those of training-time
> pruning methods. While at the time of writing the paper we felt that these approaches have a different
> motivation (i.e., model size reduction and not training speedups), we understand how these results may be
> informative to the reader.
>
> Concerning the comment on matrix-vector product complexity, we agree that lack of reuse is the problem.
> We aim to support batch size of 1 in order to allow data-parallel training on very large clusters, where
> the cumulative batch size can be in hundreds of thousands of samples. By reducing the batch size on each
> individual machine, we can keep the total batch size at a manageable level. Furthermore, if we are able
> to remove the memory bottleneck, batching is no longer necessary as a streaming architecture that keeps
> the model on-chip should have the same performance as a large-batch architecture. We will clear up these
> points in the next iteration of the paper.
>
> Again, we wish to thank the reviewer for all the effort, the provided references, and the help in
> bettering this paper.

---

### Official Review · AnonReviewer4 · 2019-10-30
**Official Blind Review #4**

**Rating:** 3

**Review:**

This paper proposes to replace dense layers with multiple sparse linear layers.  The idea is that the product ABC (for A, B, C all sparse matrices)  can accurately approximate a dense matrix D, but A(B(Cx))) requires much less work than Dx.  The paper then continues with the assumption that topology of the sparse matrices should be fixed before training, and that given this assumption we would like to find the "best" fixed topology to pick.  The paper introduces a new task to determine the "best" topology - that of reconstructing a random dense matrix.  On somewhat of a tangent the paper also introduces a minor modification to the Xavier initialization scheme that works better for deep stacks of sparse layers.

My current decision is one of Weak Reject.  I think the paper tackles an interesting line of research with some interesting ideas, but I'm concerned that the implications of the major assumptions that are made are never examined.

It is briefly implied that if the topology were fixed, then we could build hardware for such a topology (motivating the approach).  But we could also build hardware to accelerate general dynamic sparsity.  Given that it seems important to try and understand the tradeoffs involved in using fixed sparse topologies like the ones proposed and dynamic sparsity techniques such as:

1) Deep Rewiring (https://arxiv.org/abs/1711.05136)
2) Sparse Evolutionary Training (https://www.nature.com/articles/s41467-018-04316-3)
3) Dynamic Sparse Reparameterization (https://arxiv.org/abs/1902.05967)
4) Sparse Networks from Scratch (https://arxiv.org/abs/1907.04840)
5) Rigging the Lottery (https://openreview.net/forum?id=ryg7vA4tPB)

The tradeoffs both in terms of the hardware that could be built for these different regimes _and_ the effect of static topologies vs. dynamic ones on accuracy (for a given parameter / flop / energy / etc. budget).  These dynamic techniques could also be used to decompose a dense layer into a product of multiple learned sparse matrices.

The effect of assuming that we want each connection to have equal controllability seems non-obvious. For example, if we imagine that we're replacing a convolutional (or at least locally connected) layer, then we _want_ to be able to take advantage of the structure/particularities of the input and doing so will lead to a more efficient model than one which is forced to ignore them (as the proposed topology necessarily does.)   How much less efficient will the proposed architecture be in this case?

Many of these questions could be answered by trying the dynamic sparse techniques and the proposed static topologies on real problems (MB on ImageNet for example), maybe some kind of language modeling task, etc.  I find the use of one artificial task (of matrix reconstruction) which serves mainly to confirm the assumptions that are made rather than test them on real data and real problems a big weakness of the paper.

Some general notes:

The enforcing sparsity before training section should mention SNIP: Single-shot Network Pruning based on Connection Sensitivity https://arxiv.org/abs/1810.02340

The enforcing sparsity during training section, should mention both the dynamic techniques that are mentioned above, but also techniques that are dense -> sparse but which significantly outperform the L1 techniques mentioned.

For example:

1) Iterative Pruning as used in Learning both Weights and Connections for Efficient Neural Networks (https://arxiv.org/abs/1506.02626) and popularized by The Lottery Ticket Hypothesis (https://arxiv.org/abs/1803.03635)
2) To prune or not to Prune (https://arxiv.org/abs/1710.01878)
3) Variational Dropout (https://arxiv.org/abs/1701.05369)
4) Dynamic Network Surgery (https://arxiv.org/abs/1608.04493)

The mobilenet reference seems a bit out of place as mobilenets are never otherwise used in the rest of the paper.  It seems like something very similar has already been done in (https://dawn.cs.stanford.edu/2019/06/13/butterfly/).

vraiance -> variance

**Experience Assessment:**

I have published in this field for several years.

**Review Assessment: Checking Correctness Of Derivations And Theory:**

I assessed the sensibility of the derivations and theory.

**Review Assessment: Checking Correctness Of Experiments:**

I carefully checked the experiments.

**Review Assessment: Thoroughness In Paper Reading:**

I read the paper thoroughly.

---

> ### Author Response · Authors · 2019-11-15
> **Review #4 response**
>
> We thank the reviewer for the detailed and helpful comments.
>
> The reviewer mentions that we should understand the trade-offs between using static and dynamic sparsity.
> We agree and will update our paper with a performance and accuracy comparisons of applying dynamic sparsity
> during training, vs. pre-selecting a topology for static sparsity.
>
> Concerning the comment on whether we want equal controllability, the reviewer gives an example of CNNs
> where we would want to exploit local structure. Note that our method does not sparsify the network in the
> spatial domain, but only attempts to sparsify inter-channel connections. Similarly to MobileNet, we still
> use a large number of depthwise convolutions. Only the interpolation between the convolved channels is
> affected by a priori pruning. We will update the explanation to remove any ambiguities.
>
> Due to space constraints we did not add any evaluations on CIFAR10/ImageNet in the paper. We will update
> the appendix with some recent results to showcase the applicability of our method.
>
> Finally, we will integrate the reviewer's comments about the related work.
>
> Again, we thank the reviewer for their time and effort in helping us better this paper.

---

> > ### Comment · AnonReviewer4 · 2019-11-15
> > **Response**
> >
> > I didn't mean that the paper specifically suggests replacing spatial operations.  I was using spatial operations as an example of a case where it is obvious that the criterion of uniform controllability is undesirable - it is certainly not the only case.  It seems like any problem where there is structure to be exploited would have a more efficient representation with non-uniform controllability.  The chosen problem of random matrix reconstruction has no structure, so the problem and solution seem well matched, but I'm not convinced that this is generally representative.
> >
> > I look forward to seeing results on CIFAR-10 and ImageNet.

---

### Decision · Program_Chairs · 2019-12-19

**Decision:**

Reject

**Comment:**

This work proposes new initialization and layer topologies for training a priori sparse networks. Reviewers agreed that the direction is interesting and that the paper is well written. Additionally the theory presented on the toy matrix reconstruction task helped motivate the proposed approach. However, it is also necessary to validate the new approach by comparing with existing sparsity literature on standard benchmarks. I recommend resubmitting with the additional experiments suggested by the reviewers.